# Engineering of a Long-Acting Bone Morphogenetic Protein-7 by Fusion with Albumin for the Treatment of Renal Injury

**DOI:** 10.3390/pharmaceutics14071334

**Published:** 2022-06-24

**Authors:** Mei Takano, Shota Toda, Hiroshi Watanabe, Rui Fujimura, Kento Nishida, Jing Bi, Yuki Minayoshi, Masako Miyahisa, Hitoshi Maeda, Toru Maruyama

**Affiliations:** Department of Biopharmaceutics, Graduate School of Pharmaceutical Sciences, Kumamoto University, 5-1, Oe-honmachi, Chuo-ku, Kumamoto 862-0973, Japan; 195y2003@st.kumamoto-u.ac.jp (M.T.); osktd-shota@outlook.jp (S.T.); bu.bu.bu3pigs@gmail.com (R.F.); spbv8d99@gmail.com (K.N.); bijing0418@hotmail.com (J.B.); jnjymn332kg@gmail.com (Y.M.); ecoute.msk@gmail.com (M.M.); maeda-h@kumamoto-u.ac.jp (H.M.); tomaru@gpo.kumamoto-u.ac.jp (T.M.)

**Keywords:** albumin fusion, bone morphogenetic protein-7, renal fibrosis, blood retention, osteogenic activity

## Abstract

The bone morphogenetic protein-7 (BMP7) is capable of inhibiting TGF-β/Smad3 signaling, which subsequently results in protecting the kidney from renal fibrosis, but its lower blood retention and osteogenic activity are bottlenecks for its clinical application. We report herein on the fusion of carbohydrate-deficient human BMP7 and human serum albumin (HSA-BMP7) using albumin fusion technology and site-directed mutagenesis. When using mouse myoblast cells, no osteogenesis was observed in the glycosylated BMP7 derived from Chinese hamster ovary cells in the case of unglycosylated BMP7 derived from *Escherichia coli* and HSA-BMP7. On the contrary, the specific activity for the Smad1/5/8 phosphorylation of HSA-BMP7 was about 25~50-times lower than that for the glycosylated BMP7, but the phosphorylation activity of the HSA-BMP7 was retained. A pharmacokinetic profile showed that the plasma half-life of HSA-BMP7 was similar to that for HSA and was nearly 10 times longer than that of BMP7. In unilateral ureteral obstruction mice, weekly dosing of HSA-BMP7 significantly attenuated renal fibrosis, but the individual components, i.e., HSA or BMP7, did not. HSA-BMP7 also attenuated a cisplatin-induced acute kidney dysfunction model. The findings reported herein indicate that HSA-BMP7 has the potential for use in clinical applications for the treatment of renal injuries.

## 1. Introduction

Renal fibrosis is a major hallmark and a main pathological component of chronic kidney disease (CKD), regardless of the initial cause [1]. It is widely accepted that transforming growth factor-β (TGF-β) and its downstream Smad cascade are key mediators in the pathogenesis of renal disease, given its multiple signals that include renal fibrosis, inflammation, and apoptosis [2,3]. Therefore, attempts to develop therapeutic strategies that target TGF-β signaling against renal fibrosis have been reported.

TGF-β and the bone morphogenetic protein-7 (BMP7), two key members of the TGF-β superfamily, play important but diverse roles in renal fibrosis. Both TGF-β and BMP7 share similar downstream Smad signaling pathways, but they counter-regulate each other, thus allowing a balance between their biological activity to be maintained [4]. TGF-β binds to TGF-β receptors to convey intracellular signals through Smad2/3, which are pathogenic in the context of renal fibrosis. In contrast, BMP7 binds to BMP receptors to activate Smad1/5/8, which leads to renoprotective effects [4]. BMP7 is expressed at high levels in the kidney, and its genetic deletion leads to the severe impairment of kidney development [5]. On the other hand, the overexpressed renal BMP7 is capable of inhibiting TGF-β/Smad3 signaling, which protects the kidney from TGF-β-mediated renal injury [6]. This counter-regulation suggests that restoring BMP7 in CKD could have some therapeutic potential. Several reports have demonstrated that the repeated administration of human BMP7 suppressed acute and chronic kidney injuries such as ischemic acute renal failure [7], diabetic nephropathy [6,8,9], nephrotoxic nephritis [10], and Alport syndrome [11] in animal models.

However, there are two major problems associated with clinical applications of BMP7, one of which is its short half-life. Because it is a low-molecular-weight protein, BMP7 is rapidly eliminated from the body after its administration via glomerular filtration. A study by Vukicevic et al. concerning the pharmacokinetic properties of BMP7 in rats reported that the levels of BMP7 steadily declined with a half-life of ~30 min [7], which is extremely short. In addition, when BMP7 is systemically administered, only 0.5% of the administered BMP7 dose/g of tissue is available to interact with BMP7 receptors in the kidney. Because of this, for BMP7 to be clinically viable, it would need to be administered by frequently repeated administration or a huge amount of BMP7 to achieve renoprotective action in vivo. The other problem is that large doses of BMP7 might exert adverse effects such as osteogenic action, which could lead to the development of cardiovascular disease [12,13]. In fact, it was reported that the administration of BMP7 induced the ectopic formation of the bone protein in an animal model [14]. To overcome these two problems, we prepared an engineered version of BMP7. To extend the plasma half-life of BMP7, we applied albumin fusion technology, which permits the blood retention properties of protein-based pharmaceutics that have a short circulation half-life to be extended. In addition, to reduce the osteogenic activity of BMP7, three N-glycosylation sites of BMP7 were mutated (N10Q, N29Q, and N80Q) based on a previous report in which the osteogenic activity of BMP6, the closest relative to BMP7, was diminished when the carbohydrate moieties were absent [15].

The purpose of this study is to produce a fusion protein in which unglycosylated BMP7 was fused with albumin (HSA-BMP7) using a Pichia expression system that would be expected to result in a product that would be retained in the blood retention for longer periods and would also have lower osteogenic activity. The effects of HSA-BMP7 on its in vitro activity, pharmacokinetic properties as well as in vivo renoprotective activity were examined. The findings presented herein provide evidence to show that HSA-BMP7 prevents unilateral ureteral obstruction (UUO)-induced renal fibrosis and cisplatin-induced nephropathy.

## 2. Materials and Methods

### 2.1. Materials

Glycosylated BMP7 derived from Chinese hamster ovary (CHO) cells and unglycosylated BMP7 derived from *Escherichia coli* (*E. coli*) were purchased from ProSpec, Inc. (East Brunswick, NJ, USA). Human serum albumin was purchased from Kaketsuken (Kumamoto, Japan) and was defatted by Chen’s protocol [16]. TGF-β was purchased from PeproTech, Inc. (Rocky Hill, NJ, USA). Cisplatin (Randa™) was purchased from Nippon Kayaku Co., Ltd. (Tokyo, Japan). Other chemicals were of reagent grade or of the highest purity available commercially. All methods were carried out in accordance with approved guidelines. All experimental protocols were approved by Kumamoto University.

### 2.2. Production of HSA-BMP7 Fusion Protein

The genetic fusion of BMP7 and HSA was performed, and the fusion protein was produced following a previously reported method [17,18]. The details of the procedure are described in the Appendix A.

### 2.3. Circular Dichroism Spectral Measurement

CD spectra were measured using JASCO J-820 spectropolarimeter (JASCO, Tokyo, Japan) at 25 °C. The details of the procedure are described in the Appendix A.

### 2.4. In Vitro Activity of HSA-BMP7

In vitro activity of HSA-BMP7 was determined on the phosphorylation of Smad1/5/8 in human kidney (HK-2) cells, and its osteogenic activity, the alkaline phosphatase (ALP) activity in mouse myoblast cells (C2C12 cells) was also determined. The details of the procedure are described in the Appendix A.

### 2.5. Pharmacokinetic Analyses of HSA-BMP7

A 125-Iodinated fusion protein was prepared by our previously reported procedures and purified using a PD-10 column (GE Healthcare Bio-Sciences Corp., Piscataway, NJ, USA) [17]. All animal experiments were undertaken in accordance with the guideline principle and procedures of Kumamoto University for the care and use of laboratory animals (No. A 2021-021). The details of the procedure are described in the Appendix A.

### 2.6. Mouse Model of Unilateral Ureteral Obstruction (UUO) Induced Renal Fibrosis

The anti-fibrotic effect of HSA-BMP7 was evaluated in mice with unilateral ureteral obstruction (UUO). The details of the procedure are described in the Appendix A.

### 2.7. Histopathological Analysis

The details of the procedure of Masson’s trichrome staining and the immunostaining are described in the Appendix A.

### 2.8. Determination of Hydroxyproline Levels

The hydroxyproline content was determined as described previously [19]. The details of the procedure are described in the Appendix A.

### 2.9. Quantitative RT-PCR Analysis

The total RNA of kidney was extracted using RNAiso PLUS. The details of the procedure and the primer sequences are described in the Appendix A.

### 2.10. Cisplatin-Induced Acute Kidney Injury Model

Male ICR mice at 4 weeks of age were randomized to receive saline or HSA-BMP7 (200 nmol/kg) 30 min before the intraperitoneal administration of cisplatin (15 mg/kg). The details of the procedure are described in the Appendix A.

### 2.11. Biochemical Evaluation of Blood and Urine Samples

Measurements of serum creatinine (SCr), blood urea nitrogen (BUN), and urinary creatinine concentrations are described in the Appendix A.

### 2.12. Histologic Examination of Renal Tissues

The details of the histologic examination are described in the Appendix A.

### 2.13. Statistical Analyses

The means for the groups were compared by analysis of variance followed by Tukey’s multiple comparison. A probability value of *p* < 0.05 was considered to be significant.

## 3. Results

### 3.1. Structural Properties of HSA-BMP7 Fusion Protein

Figure 1A shows the diagram of the HSA-BMP7 fusion protein constructed in this study. HSA and human unglycosylated BMP7 (refer as BMP7) are genetically fused via a polyglycine and serine linker on a Pichia expression vector (see methods section). Native BMP7 contains three N-glycosylation sites (N10, N29, and N80). The three N-glycosylation sites of BMP7 were mutated (N10Q, N29Q, and N80Q) to reduce its osteogenic activity. The fusion protein was expressed and purified as HSA-unglycosylated BMP7 (referred to as HSA-BMP7) in a Pichia expression system.

The purified HSA-BMP7 was subjected to SDS-PAGE (Figure 1B) with HSA as a control. The data showed a single band with a molecular weight corresponding to those for HSA and for HSA plus unglycosylated BMP7. Western blot analysis using an anti-HSA antibody and an anti-BMP7 antibody was also performed (Figure 1B). The anti-HSA antibody produced bands for the HSA and the HSA-BMP7 fusion protein. Western blot with the anti-BMP7 antibody produced the band for only the fusion protein, further confirming that BMP7 was linked to HSA in the fusion protein. In addition, circular dichroism (CD) measurements were performed in far-UV regions to obtain information on the secondary protein structure of the purified protein (Figure 1C). Both the fusion protein and HSA showed the same minima and shape in the far-UV CD spectra, suggesting the fusion did not result in any significant change in the tertiary structure of HSA. We previously reported that a similar phenomenon was observed in the CD spectra of other HSA fusion proteins, such as HSA-thioredoxin [17].

### 3.2. In Vitro Activity of HSA-BMP7 Relative to Native Glycosylated BMP7

The BMP7 activity of the fusion protein was examined. Both TGF-β and BMP7 share similar downstream Smad signaling pathways, but they counter-regulate each other so as to maintain a balance between their biological activities [4]. First, to investigate the renoprotective activity against TGF-β-induced renal tubular toxicity, we tested the ability of the fusion protein to induce the phosphorylation of Smad1/5/8 using human kidney cells (HK-2 cells) (Figure 2A,B). BMP7, at a concentration of 10 nM, caused an increase in the phosphorylation of Smad1/5/8 compared with the control (untreated), while HSA-BMP7 showed a weaker activity than BMP7. In fact, 250 nM or 500 nM HSA-BMP7 showed similar phosphorylation activities of Smad1/5/8 with 10 nM BMP7. We next examined the effect of HSA-BMP7 on the TGB-β-stimulated expression of α-smooth muscle actin (α-SMA). As shown in Figure 2B, 10 nM BMP7 significantly suppressed the TGB-β-stimulated expression of α-SMA, and 500 nM HSA-BMP7 also significantly suppressed the expression of α-SMA as well. These data indicate that the fusion protein retains BMP7 activity with respect to Smad signaling, although the specific activity was weaker than that for the native BMP7. The osteogenic activity of HSA-BMP7 was also examined by measuring alkaline phosphatase (ALP) activity in C2C12 cells, a mouse myoblast cell line. The ALP activity in C2C12 cells has been widely used for determining the osteogenic activity of BMP7. Incubation with 15~60 nM BMP7 caused a dose-dependent increase in ALP activity, while incubation with 30 nM and 500 nM HSA-BMP7 or 30 nM unglycosylated BMP7 (from *E. coli*) did not (Figure 2C). These data suggest that the glycosylation of BMP7 appears to be necessary for its osteogenic activity. It is also possible that HSA-BMP7 could have a lower osteogenic activity than glycosylated BMP7.

### 3.3. Pharmacokinetic Profile of HSA-BMP7 in Mice

The pharmacokinetics of ^125^I-labeled HSA-BMP7, HSA, or BMP7 was evaluated in healthy mice. Each radiolabeled protein (dose: 0.1 mg/kg) was intravenously administrated to mice via the tail vein, and the plasma concentration profile of the trichloroacetic acid precipitate was determined by means of an auto-well gamma counter. The profile for the fusion protein in plasma (plasma half-life: ~10 h) was about the same as that of HSA, but BMP7 was rapidly eliminated from the blood (<30 min) (Figure 3), as has been reported in previous studies [7]. These results suggest that HSA-BMP7 has a substantially longer retention time in blood circulation compared with BMP7.

### 3.4. Effect of HSA-BMP7 on the Renal Fibrosis in Unilateral Ureteral Obstruction Mice

The anti-renal fibrotic effect of HSA-BMP7 was evaluated by using unilateral ureteral obstruction (UUO) mice. Saline, HSA-BMP7, HSA, or BMP7 (100 nmol/kg) was administered intravenously via the tail vein immediately after and at 7 days after the UUO (Figure 4A). At 14 days after the UUO, the degree of tubulointerstitial fibrosis was determined by Masson’s Trichrome-staining. As shown in Figure 4B, the administration of 100 nmol/kg of the HSA-BMP7 significantly attenuated UUO-induced renal fibrosis (28%) in comparison with the control group (50%), which was administered saline. In contrast, the administration of an equal molar amount of HSA (59%) or BMP7 (48%) had no effect on improving UUO-induced renal fibrosis in comparison with the HSA-BMP7 group. These findings indicate that the anti-fibrotic effect of HSA-BMP7 was due to the long-acting effect of BMP7. Under the same experimental conditions, the administration of HSA-BMP7 significantly suppressed an increase in α-SMA (Figure 4C) positive myofibroblast cells and hydroxyproline levels in the kidney (Figure 4D). In addition, HSA-BMP7 significantly suppressed the UUO-induced mRNA expression of collagen 1a2 (Col1a2) and α-SMA, but it had no effect on the expression of TGF-β (Figure 4E).

### 3.5. Effect of HSA-BMP7 on Cisplatin-Induced Nephropathy in Mice

The levels of BUN, SCr, and creatinine clearance (CCr) were measured at 96 h after the intraperitoneal administration of cisplatin (15 mg/kg) (Figure 5A). As shown in Figure 5B, cisplatin administration (saline group) resulted in an elevated BUN, SCr, and a decrease in CCr compared to the healthy group. The administration of a single intravenous dose of HSA-BMP7 (200 nmol/kg) at 30 min prior to the cisplatin injection significantly attenuated the cisplatin-induced renal dysfunction as compared with the saline-administered control group (Figure 5B).

Figure 5C provides information on histological alterations, as evidenced by Periodic acid-Schiff (PAS) staining and semi-quantitative scoring analysis of the kidneys of the healthy and cisplatin group with or without HSA-BMP7 administration. Because cisplatin accumulates in proximal tubular epithelial cells, the most severe and pronounced alterations occur in the renal tubule cortex and outer stripe of the outer medulla. As shown in Figure 5C, as expected, such severe alterations were observed in the renal cortico-medullary boundary zone. The cisplatin group showed evidence of tubular damage, detachment, and foamy degeneration of the tubular cells. However, the HSA-BMP7 administration significantly reduced the extent of renal tubular injury. These morphological changes are entirely consistent with the alteration in renal functions, as shown in Figure 5B.

In order to evaluate renal tubular apoptosis, terminal deoxynucleotidyl transferase-mediated deoxyuridine triphosphate nick-end labeling (TUNEL) immunostaining was performed (Figure 5D). The kidneys from the cisplatin group showed a marked increase in the number of TUNEL positive-apoptotic cells as compared with healthy mice. On the other hand, the HSA-BMP7 administration significantly reduced the number of TUNEL-positive cells as compared with the saline group.

## 4. Discussion

This study was undertaken in an attempt to extend the blood retention time and reduce the osteogenic activity of BMP7 to overcome the bottleneck associated with its clinical application. By using albumin fusion technology and site-directed mutagenesis, a fusion of carbohydrate-deficient BMP7 and HSA was prepared, and its structure, biological activity, and renoprotective effects were evaluated. As a result, we successfully produced an HSA-BMP7 fusion preparation that permitted the retention of the material in the blood to be extended, reduced osteogenic ability, and exerted an inhibitory effect against renal injury.

We first examined the biological activity of HSA-BMP7 using an in vitro system. As a result of evaluating its osteogenic potential (ALP activity) using C2C12 cells, the osteogenic activity observed in the glycosylated BMP7 derived from CHO cell was not observed in the case of the unglycosylated BMP7 derived from *E. coli* and HSA-BMP7. On the other hand, regarding the phosphorylation activity of Smad 1/5/8 in HK-2 cells, a concentration of HSA-BMP7 that was about 25~50 times higher was needed to obtain a similar signal compared with that of BMP7 derived from CHO cells, indicating that the specific activity for Smad1/5/8 phosphorylation was about 25~50-times lower, but that the phosphorylation activity of Smad1/5/8 was retained. It is known that BMPs, members of the TGF-β superfamily, exert their biological activities by inducing the formation of a heteromeric receptor complex between type II (activin receptor (ActR)-II, ActRIIB or BMPR-II) and type I (activin receptor-like kinase (Alk3/BMPR-IA), Alk6/BMPR-IB and Alk2/ActR-I) [20,21,22]. In a previous study, Saremba et al. reported that osteogenic activity could be diminished by removing the carbohydrate moiety of BMP6, the closest relative of BMP7, in which one of four types of BMP receptors, Alk2/ActR-I could be involved in osteogenic activity [15]. In fact, they revealed that the unglycosylated form of BMP6 that was expressed from *E. coli* cells did not bind to Alk2/ActR-I, whereas similar binding parameters were found for the other type I receptors Alk3/BMPR-IA and Alk6/BMPR-IB and for the type II receptor ActR-II. It, therefore, appears that the N-glycosylation plays an important role in the Alk2/ActR-I mediated signaling of BMP6, such as osteogenic activity. Sugimoto et al. also reported that a cyclic peptide (THR-123) consisting of the sequence of the non-glycosylation site of BMP7 showed a renal protective effect on renal fibrosis [23]. They also reported that the THR-123 peptide acts specifically through Alk3/BMPR-IA signaling, a type I receptor, which is important for kidney regeneration and the reversal of fibrosis. Therefore, we speculate that the carbohydrate chain of BMP7 is involved in osteogenic activity but is less involved in renal repair activity, whereas the peptide moiety of BMP7 is necessary for its anti-fibrotic effects. In other words, the results obtained from our study may be related to the differences in BMP receptors that are involved in osteogenesis and renal repair and the presence or absence of carbohydrate moieties in each receptor recognition. Further investigations would be needed to clarify the contribution of HSA-BMP7 and each BMP receptor interaction for its biological activity. In addition, it will also be necessary to evaluate bone formation ability in an in vivo animal model. Not only the presence or absence of carbohydrate moiety but also the mode of fusing with HSA might reduce the specific activity of BMP7. In this connection, a decrease in the specific activity of proteins due to macromolecularization by polyethylene glycol has been reported [24], and similar results were also reported with HSA fusing in our previous study [17]. This may be due to the spatial suppression of receptor recognition between the functional protein and its receptor by fusing with HSA, and similar phenomena might also be involved in the case of HSA-BMP7.

The plasma half-life of HSA-BMP7 was similar to that of HSA but was about 8 h higher as compared with BMP7 alone. Similar results have been obtained for an HSA-thioredoxin fusion preparation that was developed in our laboratory [17]. The fact that the half-life of HSA in humans is around 20 days suggests that the administration of HSA-BMP7 to humans would show a similar half-life. Therefore, in humans, it might be possible to administer HSA-BMP7 once a week or at two- to three-week intervals. In fact, an albumin-fused factor IX was recently developed, and it was approved for dosing at a frequency of once per week or at two-week intervals [25]. Since, in this study, the blood retention of HSA-BMP7 was only studied using healthy mice, it would be necessary to examine its pharmacokinetic profiles in animals with kidney disease.

Using UUO mice as a renal fibrosis model, a once-per-week administration of HSA-BMP7 clearly suppressed renal fibrosis. At that time, an anti-fibrotic effect was not observed in the group that had been treated with HSA alone or the BMP7 group, strongly indicating that the anti-fibrotic effect of HSA-BMP7 was due to the extended retention of BMP7 in the blood. In recent years, as a new pathogenesis concept of CKD, the importance of renal fibrosis after a repetition of an acute kidney injury (AKI) has been pointed out [26]. Therefore, the in vivo activity of HSA-BMP7 against the cisplatin-induced AKI model was also examined. A single intravenous administration of HSA-BMP7 to cisplatin-treated mice suppressed cisplatin-induced renal dysfunction and damage to kidney tissue, suggesting that the HSA-BMP7 also ameliorated the pathology of AKI. Since, based on our study, HSA-BMP7 improved both AKI and renal fibrosis, a therapeutic effect against CKD would be expected. In addition, it has been reported that BMP7 exerts an anti-fibrotic effect not only in the kidney but also in other organs such as the heart and liver [27,28]. In tumor tissue, BMP7 improved the microenvironment through the suppression of extracellular matrix formation [29]. Therefore, it is possible that HSA-BMP7 may have multi-faceted practical applications.

## 5. Conclusions

A long-acting carbohydrate-deficient BMP7 prepared by combining albumin fusion technology and site-specific mutagenesis showed a sustained renoprotective activity in the mice model of UUO-induced renal fibrosis and cisplatin-induced AKI.

## Figures and Tables

**Figure 1 pharmaceutics-14-01334-f001:**
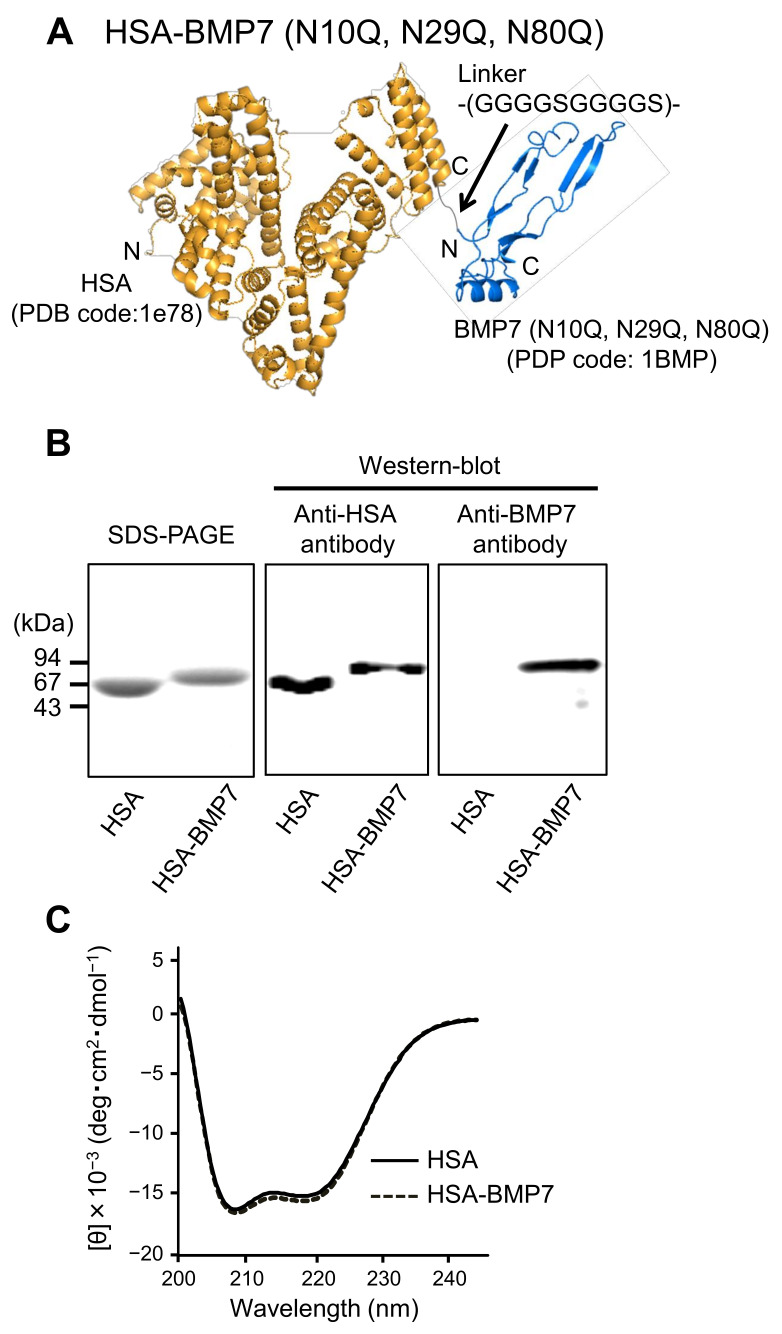
Characterization of the HSA-BMP7 fusion protein. Structural diagram (image view) of the HSA-BMP7 fusion protein containing 3 engineered point mutations at positions 10, 29 and 80 (N10Q, N29Q and N80Q) at which wild-type human BMP7 has N-glycosylated sites. (GGGGSGGGGS) indicates the polyglycine and serine linker (**A**). Reduced SDS-PAGE of HSA-BMP7 and Western blot analysis of HSA-BMP7 (**B**). The transferred PVDF membranes were incubated with primary antibodies against HSA or human BMP7. Equal amount of proteins (1 μg/lane) were electrophoresed. Far-UV circular dichroism spectra of HSA and HSA-BMP7 (**C**). The protein concentration was 2.5 μM in PBS (pH 7.4).

**Figure 2 pharmaceutics-14-01334-f002:**
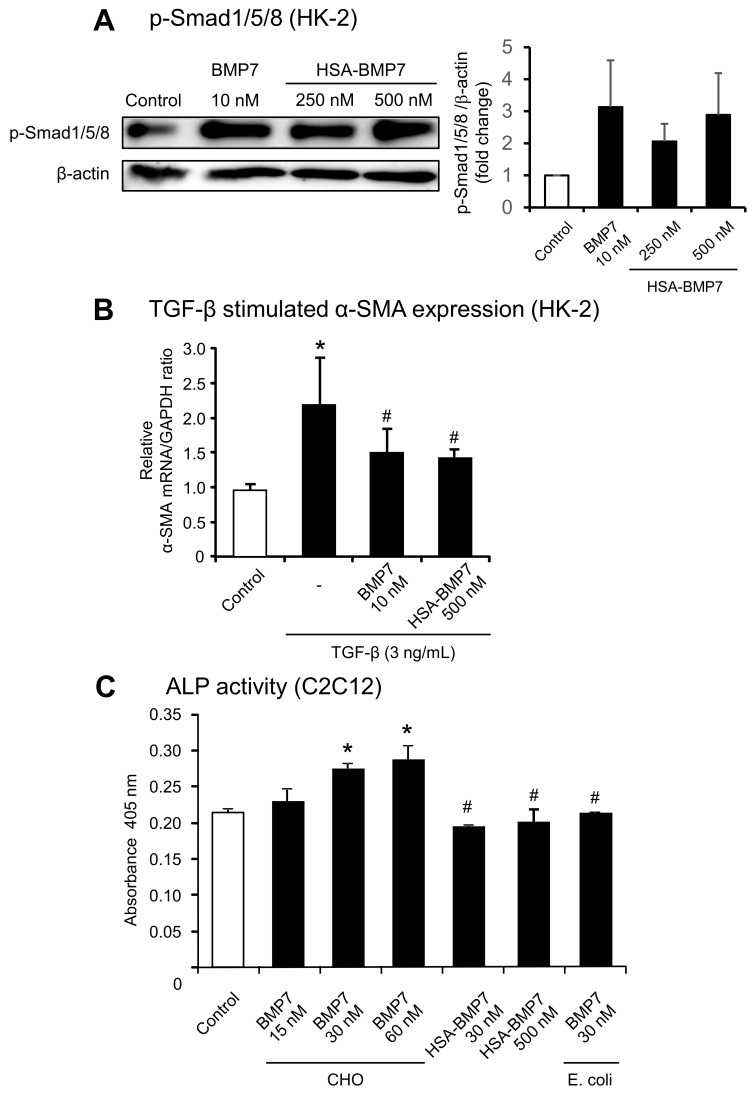
In vitro activity of HSA-BMP7 relative to native glycosylated BMP7 derived from CHO cells. Effect of HSA-BMP7 on the phosphorylation of Smad1/5/8 in HK-2 cells (**A**). HK-2 cells were incubated in 6-well plates in K-SFM medium containing 5 ng/mL of human recombinant EGF and 0.05 μg/mL of bovine pituitary extract at 37 °C for 24 h, and then incubated with BMP7 (10 nM) or HSA-BMP7 (250 nM or 500 nM) for 60 min in K-SFM. Phosphorylation of Smad1/5/8 protein HK-2 cells was determined by Western blot. Representative Western blot bands and their semi-quantitative data are shown. Results are the means ± SD (*n* = 3). Effect of HSA-BMP7 on the TGF-β stimulated mRNA expression of α-SMA in HK-2 cells (**B**). α-SMA mRNA in HK-2 cells was determined by quantitative RT-PCR in the presence of 3 ng/mL TGF-β with or without BMP7 or HSA-BMP7 for 48 h. Results are the means ± SD (*n* = 3–10). * *p* < 0.05, compared with control (untreated). # *p* < 0.05, compared with TGF-β treatment. Effect of HSA-BMP7 on alkaline phosphatase (ALP) activity in C2C12 cells (**C**). C2C12 cells were incubated in 24-well plates in DMEM with 10% heat inactivated FBS at 37 °C for 24 h. Glycosylated BMP7 derived from CHO cells, unglycosylated BMP7 derived from *E. coli*. or HSA-BMP7 was added to the DMEM with 5% FBS. The cells were incubated for 9 days. The results are the means ± SD (*n* = 3–6). * *p* < 0.05, compared with control (untreated). # *p* < 0.05, compared with 30 nM BMP7 (CHO).

**Figure 3 pharmaceutics-14-01334-f003:**
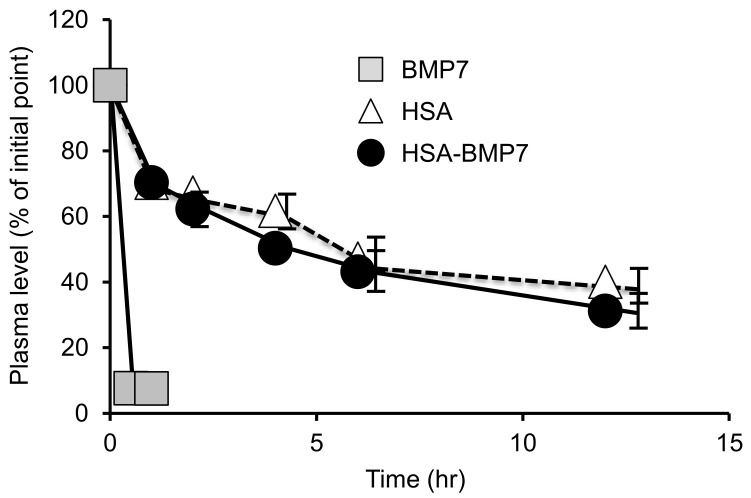
Plasma level of ^125^I-labeled HSA-BMP7, HSA or BMP7 after intravenous administration to mice. ^125^I-labeled proteins were injected through the tail vein of the mice. Results are the means ± SD (*n* = 3–4).

**Figure 4 pharmaceutics-14-01334-f004:**
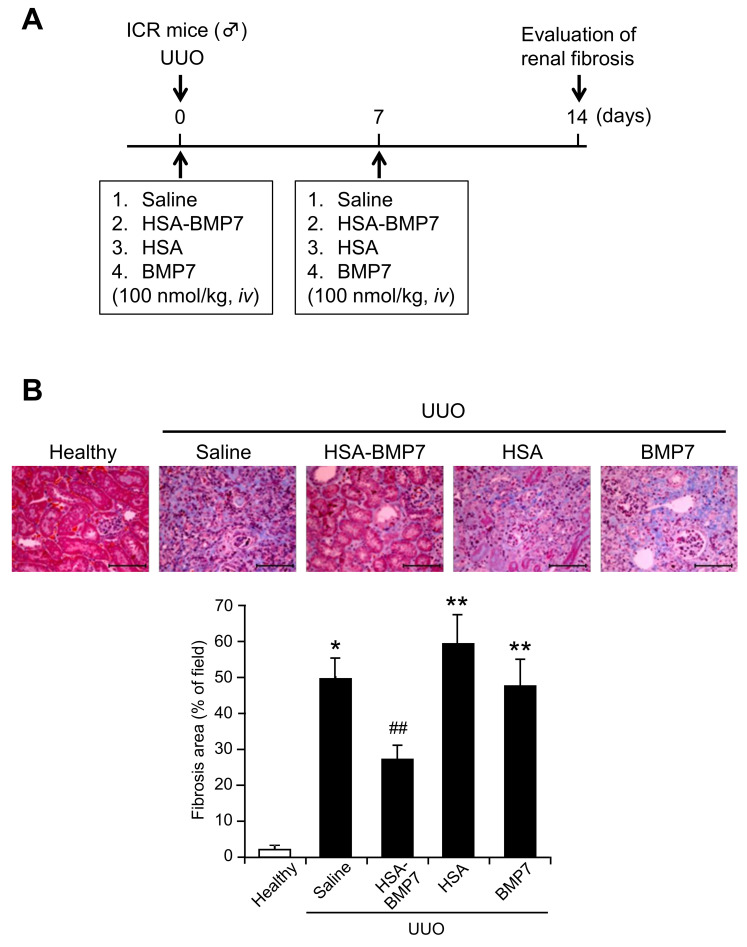
Effect of HSA-BMP7 on renal fibrosis in obstructive nephropathy. Experimental protocol for studying the effect of HSA-BMP7 against unilateral ureteral obstruction (UUO) induced renal fibrosis in mice (**A**). ICR mice underwent unilateral ligation. Two ligatures, 5 mm apart, were placed in the upper two-thirds of the ureter. Saline, HSA-BMP7, HSA or BMP7 (100 nmol/kg) was administered intravenously just after and 7 days after UUO. Masson trichrome staining of the kidneys of UUO mice with saline, HSA-BMP7, HSA or BMP7 treatment (100 nmol/kg) (**B**). Representative photomicrographs of Masson trichrome stained kidney section of healthy (non-obstructed kidney) and obstructed kidney. Results are the mean ± SD (*n* = 4–6). ** *p* < 0.05, compared with healthy mice. ## *p* < 0.05, compared with UUO with saline treatment. Immunostaining of α-SMA of the kidney of UUO mice with saline or HSA-BMP7 treatment (100 nmol/kg) (**C**). Representative photomicrographs of α-SMA-stained kidney sections of healthy (non-obstructed kidney) and obstructed kidney. Results are the mean ± SD (*n* = 4–6). * *p* < 0.05, compared with healthy mice. # *p* < 0.05, compared with UUO with saline treatment. Effect of HSA-BMP7 on hydroxyproline levels in kidney of UUO mice (**D**). Results are the means ± SD (*n* = 5–7). * *p* < 0.05, compared with healthy mice. # *p* < 0.05, compared with UUO with a saline treatment. mRNA expression of Col1a2, α-SMA or TGF-β in kidney of UUO mice with a saline or HSA-BMP7 treatment (**E**). Col1A2, α-SMA or TGF-β mRNA was determined by quantitative RT-PCR. Results are the mean ± SD (*n* = 4–7). * *p* < 0.05, compared with healthy mice. NS, not significant difference.

**Figure 5 pharmaceutics-14-01334-f005:**
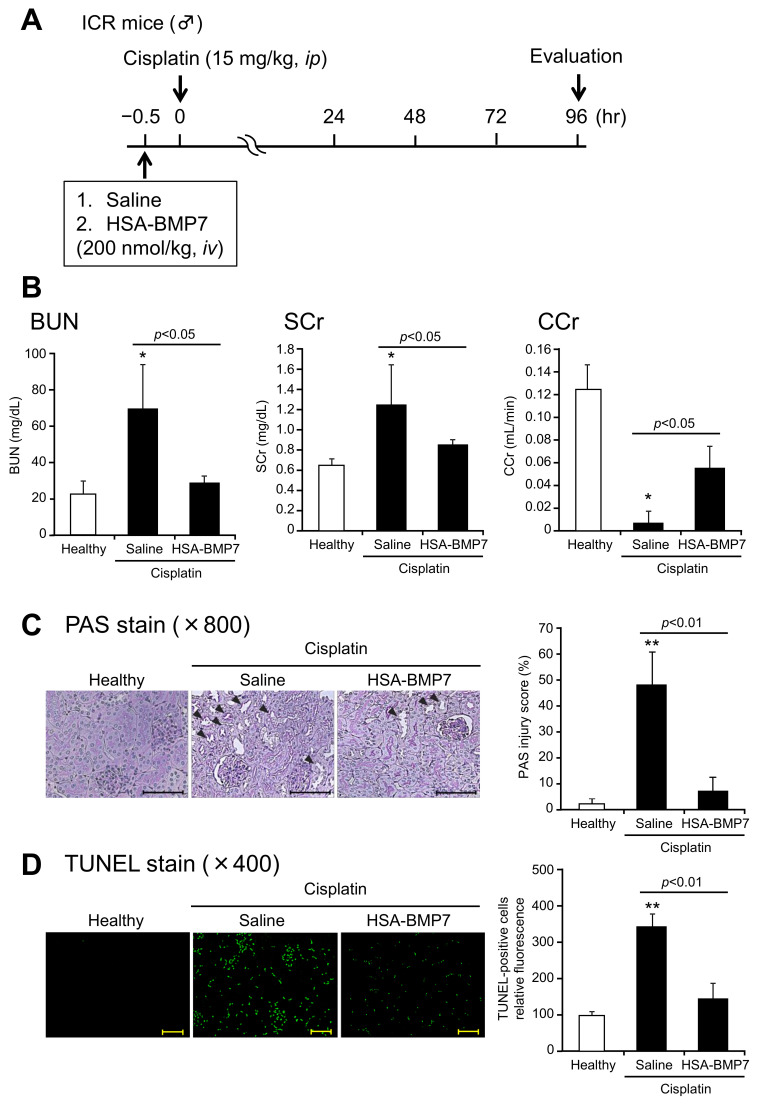
Effect of HSA-BMP7 on cisplatin-induced nephropathy. Experimental protocol for studying the effect of HSA-BMP7 against cisplatin-induced nephropathy mice (**A**). Saline or HSA-BMP7 (200 nmol/kg) was administered intravenously at 30 min before administering an intraperitoneal injection of cisplatin (15 mg/kg). Changes in the levels of blood urea nitrogen (BUN), serum creatinine (SCr) and creatinine clearance (CCr) after an intraperitoneal injection of cisplatin (15 mg/kg) (**B**). Results are the means ± SD (*n* = 4–7). * *p* < 0.05, compared with healthy mice. Histological assessment and TUNEL staining of the kidney of cisplatin-treated mice (**C**,**D**). Representative photomicrograghs of PAS and TUNEL stained kidney sections. Original magnifications: ×800 (**C**); ×400 (**D**). Scale bars represent 100 μm. The injured areas are indicated by arrows. The semi-quantitative analyses were also performed. Results are the mean ± SD (*n* = 4–7). ** *p* < 0.05, compared with healthy mice.

## Data Availability

Not applicable.

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
