# Peer review of "Engineering of a Long-Acting Bone Morphogenetic Protein-7 by Fusion with Albumin for the Treatment of Renal Injury"

_pharmaceutics, 2022, doi:10.3390/pharmaceutics14071334_

Round 1

Reviewer 1 Report

The manuscript “Engineering of long-acting bone morphogenetic protein 7 by fusion with albumin for the treatment of renal injury” by Mei Takano with co-authors demonstrates that fusion of BMP7 to the human serum albumin significantly increases protein stability in the mouse model but reduces BMP7 activity. HSA-BMP7 inhibits TGF-β -induced epithelial-to-mesenchymal (ETM) transition by phosphorylation of Smad1/5/8 and inhibition of smooth muscle actin–α (α-SMA) transcription. Additionally to counteracting of TGF-β function, BMP7 induces osteogenesis. Authors further demonstrated that mutation of three N-glycosylation sites significantly reduced osteogenesis. Mutated BMP7 and fused mutated HSA-BMP7 protein do not increase alkaline phosphatase activity in mouse myoblast cells. EMT induces accumulation of myofibroblasts and tubular injury leading to development of renal fibrosis. Administration of HSA-BMP7 prevents development of renal fibrosis in the unilateral ureteral obstruction (UUO) model and improve renal function in the mouse model of cisplatin-induced acute kidney injury.  

Overall, the manuscript is timely, well organized, and of interest for the readers. Unfortunately, it is required an intensive English spell check. Lot of sentences are too long and do not have either subject or a verb.

Specific comments:

Title. Correct “born morphogenic protein” to “bone morphogenic protein” Also correct the same misspelling in the manuscript text.

Fig. 1 Change the figure legend to Characterization of fusion protein.

Fig. 2A. Quantify the difference in the level of phosphorylation of Smad 1/5/8 detected by western blot. Why phosphorylation of Smad and α-SMA expression were tested in HK-2 cells, but ALP activity was tested in C2C12 cells? Explain in the text

Fig 5C -Bad quality of photos. Difficult to see injury. Improve quality, add arrows to point injury. Is reduction of renal injury significant after HAS-BMP7 administration?

Fig 5D. The absence of nuclear staining makes it difficult to compare apoptosis levels. Is reduction of renal injury significant after HAS-BMP7 administration? What does relative fluorescence mean? What did you use for normalization of the apoptosis levels?

Minor comments

Line 19-22 The sentence should be revised: Using mouse myoblast cells assay, the osteogenic activity was observed for glycosylated BMP7, isolated from hamster ovary, but not for non-glycosylated recombinant BMP7 or HSA-BMP7, produces in E.coli. Replace “osteogenic activity” to “osteogenesis”.

Line 66 Explain what is a “born protein” mean or change it to  “bone protein.”

Line 66 ” To overcome these two problems, we… “ move this information to the last paragraph of abstract. Place information about the published report that mutated BMP6  reduce osteogenesis before description of your approach in this report.

Line 84, 138, 144, 162 – “BMP7 derived from…” replace to “BMP7 isolated from…”

Line 135, 138, 160 Provide full name for HK-2, CHO and C2C12 cells

Line 142 Provide full name for α-SMA

Line 209 “by mean of HistoVT One.” replace to “using HistoVT One.”

Lines 211, 213 replace “kidney slices” to “kidney sections”

Line 230 Please provide a full name of Col1a2

Line 270 The “image diagram” remove “image”

Line 303 “First, the activity for phosphorylation of Smad1/5/8…” re-write like “First, we tested an ability of the fusion protein to induce phosphorylation of Smad1/5/8.”

Line 318 – Correct “Figure 4C” to “Figure 2C” The levels are below control?

Line 450 “BMP6 that was expressed from E.coli…” do you mean mutated BMP6?

Author Response

Comment-1

It is required an intensive English spell check. Lot of sentences are too long and do not have either subject or a verb.

Reply-1

We asked Dr. Milton S, Feather to perform a second examination of our revised paper for English usage and he has done so. Dr. Feather, a retired professor of Biochemistry (University of Missouri, USA) operates TECHNICAL EDITING SERVICES, an editing service that specializes in examining, editing and correcting scientific manuscripts that have been written in English, by foreign scientists and which are intended for publication in an English-language Journal. A letter of confirmation regarding this paper from Dr. Feather is attached.

Comment-2

Title. Correct “born morphogenic protein” to “bone morphogenic protein” Also correct the same misspelling in the manuscript text.

Reply-2

As the reviewer pointed out, the word “born” was corrected to “bone” in the revised manuscript.

(title, abstract, keywords and introduction)

Comment-3

Fig. 1 Change the figure legend to Characterization of fusion protein. 

Reply-3

As the reviewer suggested, the figure legend in Figure 1 “Structural properties of the HSA-BMP7 fusion protein.” was changed to “Characterization of the HSA-BMP7 fusion protein” in the revised manuscript. (Figure 1)

Comment-4

Fig. 2A. Quantify the difference in the level of phosphorylation of Smad 1/5/8 detected by western blot. Why phosphorylation of Smad and α-SMA expression were tested in HK-2 cells, but ALP activity was tested in C2C12 cells? Explain in the text.

Reply-4

As the reviewer suggested, the level of phosphorylation of Smad 1/5/8 was quantified and the results are shown in Figure 2A. The following sentences were added in the legend for Figure 2 in the revised manuscript. “Representative Western blot bands and their semi-quantitative data are shown. Results are the means ± SD (n = 3).”

 First, to investigate the renoprotective activity against TGF-b-induced renal tubular toxicity, we tested the ability of the fusion protein to induce the phosphorylation of Smad1/5/8 using human kidney cells (HK-2 cells) (Figure 2A and B). On the other hand, the osteogenic activity (adverse effect) of HSA-BMP7 was also examined by measuring alkaline phosphatase (ALP) activity in C2C12 cells, a mouse myoblast cell line. The ALP activity in C2C12 cells has been widely used for determining the osteogenic activity of BMP7.

  Therefore, the following sentence in the Results section of the original manuscript, “First, the activity for the phosphorylation of Smad1/5/8 of HSA-BMP7 was determined in human kidney cells (HK-2 cells) (Figure 2A and B).” was changed to “First, to investigate the renoprotective activity against TGF-b-induced renal tubular toxicity, we tested the ability of the fusion protein to induce the phosphorylation of Smad1/5/8 using human kidney cells (HK-2 cells) (Figure 2A and B)”. (page 5, line 517)

In addition, the following sentence “The ALP activity in C2C12 cells has been widely used for determining the osteogenic activity of BMP7.” was added in the revised manuscript. (page 6, line 537)

Comment-5

Fig 5C -Bad quality of photos. Difficult to see injury. Improve quality, add arrows to point injury. Is reduction of renal injury significant after HAS-BMP7 administration?

Reply-5

As the reviewer pointed out, Figure 5C was improved and the enlarged photos were inserted into the revised manuscript. Arrows were added in the injured area. (Figure 5C) Then, the following sentences were added in the Legend for Figure 5 of the revised manuscript.

“Representative photomicrograghs of PAS and TUNEL stained kidney sections. Original magnifications: ×800 (C); ×400 (C). Scale bars represent 100 μm. The injured areas are indicated by arrows. The semi-quantitative analyses were also performed. Results are the mean ± SD (n = 4-7) compared with healthy mice.” (Legend for Figure 5)

   In addition, as mentioned in the Results section of the revised manuscript, the HSA-BMP7 administration significantly reduced the extent of renal tubular injury. “The cisplatin group showed evidence of tubular damage, detachment and the foamy degeneration of the tubular cells. But, the HSA-BMP7 administration significantly reduced the extent of renal tubular injury. These morphological changes are entirely consistent with the alteration in renal functions as shown in Figure 5B.” (page 11, line 631)

Comment-6

Fig 5D. The absence of nuclear staining makes it difficult to compare apoptosis levels. Is reduction of renal injury significant after HAS-BMP7 administration? What does relative fluorescence mean? What did you use for normalization of the apoptosis levels?

Reply-6

In Figure 5D, we did not check the nuclear staining such as DAPI, but PAS staining was shown in Figure 5C. As mentioned in the Results section, the cisplatin group showed evidence of tubular damage, detachment and the foamy degeneration of the tubular cells. Therefore, the number of nuclei were reduced in the cisplatin treatment group. TUNEL positive cells (relative fluorescence) means relative fluorescence as compared with healthy mice (healthy vs cisplatin+saline vs cisplatin+HSA-BMP7). The kidneys from the cisplatin group showed a marked increase in the number of TUNEL positive-apoptotic cells as compared with healthy mice. On the other hand, the HSA-BMP7 administration significantly reduced the number of TUNEL positive cells as compared with the saline group.

  Therefore, the following sentences, “The kidneys from the cisplatin group showed a marked increase in the number of TUNEL positive-apoptotic cells. On the other hand, the HSA-BMP7 administration significantly reduced the number of TUNEL positive cells as compared with the saline group.” in the Results section of the original manuscript were changed to “The kidneys from the cisplatin group showed a marked increase in the number of TUNEL positive-apoptotic cells as compared with healthy mice. On the other hand, the HSA-BMP7 administration significantly reduced the number of TUNEL positive cells as compared with the saline group.” in the revised manuscript. (page 11, line 637)

Comment-7

Line 19-22 The sentence should be revised: Using mouse myoblast cells assay, the osteogenic activity was observed for glycosylated BMP7, isolated from hamster ovary, but not for non-glycosylated recombinant BMP7 or HSA-BMP7, produces in E.coli. Replace “osteogenic activity” to “osteogenesis”.

Reply-7

As the reviewer suggested the phrase “osteogenic activity” was changed to “osteogenesis” (page 1, line 19)

Comment-8

Line 66 Explain what is a “born protein” mean or change it to “bone protein.”

Reply-8

As the reviewer pointed out, the word “born” was corrected to “bone” in the revised manuscript.

(title, abstract, keyword and introduction)

Comment-9

Line 66 ”To overcome these two problems, we… “ move this information to the last paragraph of abstract. Place information about the published report that mutated BMP6 reduce osteogenesis before description of your approach in this report.

Reply-9

I was not able to understand the suggestion to “move this information to the last paragraph of abstract”. The location of the following sentence “To overcome these two problems, we… “ seems to be appropriate. In addition, the information concerning the published report that mutated BMP6 reduces osteogenesis was added in the Introduction section of the revised manuscript as follows; “In addition, to reduce the osteogenic activity of BMP7, three N-glycosylation sites of BMP7 were mutated (N10Q, N29Q and N80Q) based on a previous report in which the osteogenic activity of BMP6, the closest relative to BMP7, was diminished when the carbohydrate moieties were absent[15].” (page 2, line 83)

Comment-10

Line 84, 138, 144, 162 – “BMP7 derived from…” replace to “BMP7 isolated from…”

Reply-10

The BMP7 was secreted from CHO cells or E. Coli, then purified from the medium. We therefore feel that the term “isolated from” seems to be inappropriate.

Comment-11

Line 135, 138, 160 Provide full name for HK-2, CHO and C2C12 cells

Reply-11

As the reviewer suggested, the full names such as human kidney (HK-2) cells, chinese hamster ovary (CHO) cells, Escherichia coli (E. Coli) and mouse myoblast (C2C12) cells were provided in the revised manuscript. (page 3, line 121) , (page 2, line 97), (page 2, line 98), (page 3, line 122)

Comment-11

Line 142 Provide full name for α-SMA

Reply-11

As the reviewer suggested, the full name of a-SMA, a-smooth muscle actin, is now provided. (page 6, line 531)

Comment-12

Line 209 “by mean of HistoVT One.” replace to “using HistoVT One.”

Reply-12

As the reviewer suggested, the phrase “by mean of HistoVT One” was changed to “using HistoVT One” in the revised manuscript. (Supplemental methods: page 6, line 17)

Comment-13

Lines 211, 213 replace “kidney slices” to “kidney sections”

Reply-13

As the reviewer suggested, the phrase “kidney slices” was changed to “kidney sections” in the revised manuscript. (Supplemental methods: page 6, line 19-21)

Comment-14

Line 230 Please provide a full name of Col1a2

Reply-14

As the reviewer suggested, the full name “Collagen 1a2” of Col1a2 was added in the revised manuscript. (page 9, line 593)

Comment-15

Line 270 The “image diagram” remove “image”

Reply-15

As the reviewer suggested, the phrase “image diagram” was changed to “diagram” in the revised manuscript. (page 4, line 447)

Comment-16

Line 303 “First, the activity for phosphorylation of Smad1/5/8…” re-write like “First, we tested an ability of the fusion protein to induce phosphorylation of Smad1/5/8.”

Reply-16

As the reviewer suggested, the following sentence “First, the activity for the phosphorylation of Smad1/5/8 of HSA-BMP7 was determined in human kidney cells (HK-2 cells) (Figure 2A and B).” in the original manuscript was changed to “First, to investigate the renoprotective activity against TGF--induced renal tubular toxicity, we tested the ability of the fusion protein to induce the phosphorylation of Smad1/5/8 using human kidney cells (HK-2 cells) (Figure 2A and B).” in the revised manuscript. (page 5, line 517)

Comment-17

Line 318 – Correct “Figure 4C” to “Figure 2C” The levels are below control?

Reply-17

As the reviewer pointed out, the phrase “Figure 4C” was corrected to “Figure 2C” in the revised manuscript. (page 6, line 541) The levels are not significantly low as compared with the control values.

Comment-18

Line 450 “BMP6 that was expressed from E.coli…” do you mean mutated BMP6?

Reply-18

“BMP6 that was expressed from E.coli…” means unglycosylated BMP6 because the recombinant protein that is derived from an E. Coli. expression system is the unglycosylated form. Therefore, “the unglycosylated” was added in the revised manuscript. (page 13, line 670)

 Therefore, the following sentence “In fact, they revealed that the BMP6 that was expressed from E. coli cells did not bind to Alk2/ActR-I” in the original manuscript was changed to “In fact, they revealed that the unglycosylated BMP6 that was expressed from E. coli cells did not bind to Alk2/ActR-I” in the revised manuscript. (page 13, line 670)

Reviewer 2 Report

This is such as a complete study. Only a few minor comments:

Check the grammatical and typo errors throughout MS.

Title: Check the spelling

Abstract:

 born morphogenetic protein-7 (BMP7): Check spelling 

Introduction

 factor- (TGF-) and its downstream Smad cascade is: grammar

Few grammar mistakes, rest of the sections were written well.

2. Materials and Methods

procedures of Kumamoto University for the care and use of laboratory animals: Ethical approval No.

5. Conclusions

I would expect little more detailed conclusion.

Author Response

Comment-1

Check the grammatical and typo errors throughout MS.

Reply-1

We asked Dr. Milton S, Feather to perform a second examination of our revised paper for English usage and he has done so. Dr. Feather, a retired professor of Biochemistry (University of Missouri, USA) operates TECHNICAL EDITING SERVICES, an editing service that specializes in examining, editing and correcting scientific manuscripts that have been written in English, by foreign scientists and which are intended for publication in an English-language Journal. A letter of confirmation regarding this paper from Dr. Feather is attached.

Comment-2

Abstract: born morphogenetic protein-7 (BMP7): Check spelling

Reply-2

As the reviewer pointed out, the word “born” was corrected to “bone” in the revised manuscript.

(title, abstract, keyword and introduction)

Comment-3

Introduction: factor-b (TGF-b) and its downstream Smad cascade is: grammar

Few grammar mistakes, rest of the sections were written well.

Reply-3

As the reviewer pointed out, the following sentence “It is widely accepted that transforming growth factor- b (TGF-b) and its downstream Smad cascade is a key mediator in the pathogenesis of renal disease” in the Introduction of the original manuscript was corrected to “It is widely accepted that transforming growth factor- b (TGF-b) and its downstream Smad cascade are the key mediators in the pathogenesis of renal disease” in the revised manuscript. (page 1, line 34)

Comment-4

  1. Materials and Methods

procedures of Kumamoto University for the care and use of laboratory animals: Ethical approval No.

Reply-4

As the reviewer pointed out, Ethical approval No. was added in the revised manuscript as follows; “All animal experiments were undertaken in accordance with the guideline principle and procedures of Kumamoto University for the care and use of laboratory animals (No. A 2021- 021).” (page 3, line 126)

Comment-5

I would expect little more detailed conclusion.

Reply-5

As the reviewer suggested, the conclusion “A long-acting carbohydrate-deficient BMP7 prepared by combining albumin fusion technology and site-specific mutagenesis showed a sustained renoprotective activity.” in the original manuscript was changed to “A long-acting carbohydrate-deficient BMP7 prepared by combining albumin fusion technology and site-specific mutagenesis showed a sustained renoprotective activity in the mice model of UUO-induced renal fibrosis and cisplatin-induced AKI.” in the revised manuscript. (page 14, line 737)
